# The Biomedical Uses of Inositols: A Nutraceutical Approach to Metabolic Dysfunction in Aging and Neurodegenerative Diseases

**DOI:** 10.3390/biomedicines8090295

**Published:** 2020-08-20

**Authors:** Antonio J. López-Gambero, Carlos Sanjuan, Pedro Jesús Serrano-Castro, Juan Suárez, Fernando Rodríguez de Fonseca

**Affiliations:** 1Departamento de Biología Celular, Genética y Fisiología, Campus de Teatinos s/n, Universidad de Málaga, Andalucia Tech, 29071 Málaga, Spain; antonio.lopez@ibima.eu; 2UGC Salud Mental, Instituto de Investigación Biomédica de Málaga (IBIMA), Hospital Universitario Regional de Málaga, 29010 Málaga, Spain; 3EURONUTRA S.L., 29590 Málaga, Spain; euronutra@euronutra.eu; 4UGC Neurología, Instituto de Investigación Biomédica de Málaga (IBIMA), Hospital Universitario Regional de Málaga, 29010 Málaga, Spain; pedro.serrano.c@gmail.com

**Keywords:** Alzheimer’s disease, psychiatric disease, depression, anxiety, Down’s syndrome, inositol, nutraceutical, insulin signaling, antioxidant, aging

## Abstract

Inositols are sugar-like compounds that are widely distributed in nature and are a part of membrane molecules, participating as second messengers in several cell-signaling processes. Isolation and characterization of inositol phosphoglycans containing myo- or d-chiro-inositol have been milestones for understanding the physiological regulation of insulin signaling. Other functions of inositols have been derived from the existence of multiple stereoisomers, which may confer antioxidant properties. In the brain, fluctuation of inositols in extracellular and intracellular compartments regulates neuronal and glial activity. Myo-inositol imbalance is observed in psychiatric diseases and its use shows efficacy for treatment of depression, anxiety, and compulsive disorders. Epi- and scyllo-inositol isomers are capable of stabilizing non-toxic forms of β-amyloid proteins, which are characteristic of Alzheimer’s disease and cognitive dementia in Down’s syndrome, both associated with brain insulin resistance. However, uncertainties of the intrinsic mechanisms of inositols regarding their biology are still unsolved. This work presents a critical review of inositol actions on insulin signaling, oxidative stress, and endothelial dysfunction, and its potential for either preventing or delaying cognitive impairment in aging and neurodegenerative diseases. The biomedical uses of inositols may represent a paradigm in the industrial approach perspective, which has generated growing interest for two decades, accompanied by clinical trials for Alzheimer’s disease.

## 1. Introduction: Human Brain Aging and Inositols

The aging process in humans is associated with physical decline and impairment of metabolic homeostasis [1]. The dysregulation of the metabolic network leads to an age-related elevated risk of suffering from chronic metabolic disorders, especially insulin resistance-related pathologies. In addition to the well-known peripheral role of insulin on glucose and energy storage, insulin also regulates a series of cognitive processes, such as memory formation, through its effects on glial–neuronal metabolic coupling. Central insulin resistance is a common feature linked to premature aging and is observed in neurological disorders, including early stages of Alzheimer’s disease (AD) and Down’s syndrome (DS) [2].

Currently, 16% of the EU population is over 65, and this figure is expected to rise to 25% by 2030. Taking this trend into consideration and the prevalence of dementia, including AD, the World Health Organization estimates that population aging will lead to a dramatic increase in dementia prevalence. By 2050, more than 131.5 million people are expected to be affected. AD leads to a loss of memory and neurodegenerative cognitive functions and affects 10% of the population aged over 65 years. Delaying a cognitive decline in AD is a major research challenge and a clinical need, considering the incidence of this disease in the elderly. Common features of AD are the aggregation of β-amyloid (Aβ) plaques and tau protein hyperphosphorylation, leading to neural damage. An approach for slowing down the progress of the disease is targeting the factors that might accelerate neural damage. Present results suggest that unhealthy dietary habits, microbiota changes, and oxidative stress favor the development of brain insulin resistance, which could contribute to a neuroinflammatory profile, directly activating both the resident immune cells of the brain (microglia) and astrocytes, promoting an adverse environment for neuronal survival in the context of AD [3,4]. Accordingly, a more detailed in-depth analysis of central insulin resistance contribution to cognitive impairment is discussed later in this review. A relevant issue on the clinical approach to AD and related pathologies that lead to cognitive impairment is the fact that most of the research efforts on therapeutics have focused on either fighting the symptoms by boosting certain deteriorated transmission pathways (e.g., anti-acetylcholinesterase drugs to enhance cholinergic transmission) or reducing Aβ load via immunotherapy. However, there is a clear lack of therapeutic development designed to restore metabolic impairments associated with these neurodegenerative disorders.

The lack of a “metabolic approach to AD therapeutics” might offer an opportunity to inositols, since in the past years they have gained close attention regarding treatment of pathologies associated with altered insulin signaling. Inositols are sugar-like cyclic alcohols constituent of cells, which are normally incorporated as part of the human diet. Given their structure, there are at least eight isomers of inositols that occur in nature (*myo-*, *muco-*, *neo-*, *scyllo-*, *l-chiro-*, *d-chiro-*, *epi-*, and *allo-*inositol) and one non-occurring in nature (*cis-*inositol) (Figure 1A) [5]. Inositols act as second messengers of the insulin-signaling pathway and their administration exerts insulin-sensitizing and mimetic effects, lowering blood glucose and promoting hepatic glycogen synthesis. d-chiro-inositol has been widely used as a treatment for pathologies associated with insulin resistance, e.g., polycystic ovary syndrome (PCOS) and diabetes [6,7]. Given their polar structure, other inositols show different properties, such as scyllo-inositol, which stabilizes soluble Aβ oligomers and is being tested under clinical trials as a promising therapy for AD [8]. The use of inositols for medical purposes is closely related to their “nutraceutical” nature, although the definition of the term is still debated. Since inositols are acquired through the diet, inositol extracts can be considered a nutraceutical under the definition of an isolated or purified product from natural sources, with specific health benefits against diseases or medical conditions or a protective effect against chronic diseases. Hence, these natural compounds arise as alternatives to treatments for central and peripheral insulin resistance-related disorders.

In the present review, we provide a short description of the structure and pharmacology of inositols. However, it is not the scope of this review to describe the particular chemical characteristics of inositols or to compile their application for metabolic disorders in peripheral tissues, since several works have already elegantly described these concepts previously [5,9,10,11]. The further sections herein try to establish a descriptive line, detailing the importance of inositols and their derivatives, such as inositol-(phospho)glycans (IPGs or simply IGs) in physiological processes, highlighting their role in insulin signaling, as well as their function in the central nervous system and the perspective of their use in the treatment of neurodegenerative diseases, with a special emphasis on AD and behavioral disorders here.

## 2. Inositols in Organisms

### 2.1. Structure of Inositol Isomers and Inositol Phospholipids

Inositols are naturally occurring substances that resemble simple sugars. They have a cyclic structure of six carbons and six alcohols, being classified as sugar alcohols (polyols with one hydroxyl group attached to each carbon atom). In addition, both inositols and methyl-derived inositols are also classified as cyclitols (cycloalkanes containing at least three hydroxyl groups attached to the carbon atoms).

Thus far, the role of inositols on the body is both structural, as constituents of complex phospholipids in plasma membrane, and functional, since they act on metabolic pathways as second messengers of insulin signaling. Regarding their structural function, in mammalian tissues, inositols are found in the external part of the membrane of phosphoglycerides. The attachment of sugar residues to inositols constitutes the glycosylphosphatidylinositols (GPIs), which are elements of the outer leaflet of the plasma membrane that may serve as anchors for extracellular proteins attached to the membrane. As second messengers, inositols are a well-known part of signal transduction, as they form phosphatidylinositols (PIs) and their phosphorylated forms, phosphoinositides (PIPs) and inositol phosphates (IPs), which are responsible for membrane trafficking and cell signaling as substrates for other protein kinases. The most important members of this family are phosphatidylinositol (4,5)-bisphosphate (PI(4,5)P_2_, simplified as PIP_2_), a substrate for the phosphoinositide 3-kinase (PI3K), whose phosphorylation and conversion into phosphatidylinositol (3,4,5)-trisphosphate (PI(3,4,5)P_3_, simplified as PIP_3_) is a key step in the insulin/IGF-1 pathway. PIP_2_ is also a substrate for phospholipase C and the formation of inositol 1,4,5-trisphosphate (IP_3_), a second messenger mainly involved in intracellular Ca^2+^ trafficking in the endoplasmic reticulum.

### 2.2. Inositol Incorporation in Organisms

Despite eight out of the nine inositol isomers occurring naturally, only myo-inositol (MI; *cis*-1,2,3,5-*trans*-4,6-cyclohexanehexol), scyllo-inositol (SI; *cis*-1,3,5-*trans*-2,4,6-cyclohexanehexol), and d-chiro-inositol (DCI; *cis*-1,2,4*-trans*-3,5,6-cyclohexanehexol) have been detected as major inositols present in mammalian tissue. Some studies have reported the possible presence of neo-inositol (NI; *cis*-1,2,3-*trans*-4-5-6-cyclohexanehexol) in bovine brains [12] and also epi-inositol (EI; *cis*-1-*trans*-2,3,4,5,6-cyclohexanehexol) and muco-inositol (*cis*-1,4-*trans*-2,3,5,6-cyclohexanehexol) in the liver, muscle, blood, brain, and several other rat tissues with a rate of isomerization from MI of 0.6% [13]. MI interconverts to other inositols via an specific epimerase [13,14] that differs from the epimerase catalyzing the conversion from MI to DCI [15,16]. MI and SI are the most abundant forms of inositols in organisms, representing over 90% of the total inositol content of mammalian cells. SI has been reported to be incorporated into lipids in plants [17]. However, SI is not incorporated into PI lipids at detectable levels, even under SI administration in mammalian tissue [18]. The other inositol found in great levels in mammals is DCI, which acts as a second messenger for insulin signaling, presumably as IG, as will be depicted later. A summary of the current knowledge on inositol stereoisomers sources, distribution, and pharmacological properties can be found in Table 1.

Inositols can be obtained from the diet as they are abundant components of the cell membrane in plants and animals. The most abundant form of MI incorporation is either free, or in the form of (myo)-inositol hexaphosphate or IP_6_ (1,2,3,4,5,6-hexayl hexakis (dihydrogen (phosphate))), also known as phytic acid. Major sources of IP_6_ are plants, as IP_6_ is a major reservoir of phosphorus, energy, and a source of cations and MI in the cell wall. However, IP_6_ cannot be obtained from dietetic sources because its bioavailability is very limited. This is due to its negative charge density, being necessary to be dephosphorylated via bacterial phytases and phosphatases, providing free MI or other IPs before entering bloodstream [29,30]. The only animals that can carry out this transformation belong to the ruminant mammal group. Western diets can provide around 0.5–1 g/day of MI [31,32]. DCI is mainly obtained as the methylated form d-pinitol (DPIN; 3-o-methyl-d-chiro-inositol), which is demethylated to DCI under acidic conditions in the gastrointestinal tract. DPIN acts as an osmolyte in plants, allowing tolerance to heat, high salinity, and drought stress [33]. The *Leguminosae* family is a major source of DPIN, especially carob pods, which provide 10–80 g/kg of DPIN [34]. Many herbal extracts contain methyl-inositol derivatives like sequoyitol (5-o-methyl-myo-inositol) or l-quebrachitol (2-o-methyl-l-chiro-inositol) (Figure 1B) [35,36]. Dietary supply of other inositols seems to be scarce, as evidenced by their low presence in organisms.

Although inositols can be obtained from diet, the main inositol isomer, MI, is also synthesized in the body in great enough quantities for the whole supply required, as kidneys produce 2 g/day each of MI (4 g total) and other tissues contribute to a small extent to MI synthesis, like the brain and testis [37]. De novo MI synthesis occurs primarily from glucose into cytosol. The process follows glucose phosphorylation into glucose-6-phosphate (G6P) via hexokinase. D-3-*myo*-inositol-phosphate synthase (MIPS) catalyzes the conversion from G6P into myo-inositol 3-phosphate (MIP), and dephosphorylation occurs with inositol monophosphatase (IMPase), rendering free MI. MIPS is a phosphoprotein whose activity is regulated by the glycogen synthase kinase 3 (GSK3) homolog MCK1 in yeast [38,39]. Although this may suggest GSK3 regulation of MIPS activity in humans, there are no data of this proposed mechanism in mammalian cells. Expression of inositol-3-phosphate synthase 1 (Isyna1), a gene encoder for IMPase transcription, is tightly regulated by the disposal of phosphatidic acid (PA) and inositol hexakisphosphate synthase 1 (IP6K1). PA is considered a “metabolic sensor”, as its synthesis is upregulated by high levels of glucose or the presence of growth factors and promotes cell growth via a mammalian target of rapamycin (mTOR) [40,41]. Interaction of PA with IP6K1 leads to its translocation into the nucleus and synthesis of IP7, which negatively regulates *Isyna1* via DNA methylation [42].

The transport of inositol isomers into cells is regulated by the sodium-myo-inositol co-transporters (SMIT1 and SMIT2) with a 2 Na^+^ to 1 myo-inositol stoichiometric ratio, similar to that of the sodium-glucose co-transporter SGLT1 [43,44,45]. SMIT1 and SMIT2 both have a different affinity for inositol isomers, which are competitors for inositol transport, and are differentially regulated by monosaccharides. DCI and MI are both effectively transported into cells via SMIT2 with a similar affinity, whereas DPIN, a methyl derivative of DCI, binds to SMIT2 with lower affinity and is a competitor for MI/DCI transport [46]. SI has also a high affinity for the SMIT1 and SMIT2 transporters, whereas other inositol derivatives like sequoyitol, D-ononitol (4-o-methyl-d-chiro-inositol), or viburnitol (1-deoxy-l-chiro-inositol) are also transported with lower affinity [47]. Inositol transport is inhibited in the presence of l-fructose and d-glucose, as they are also competitors of SMIT1 and SMIT2 [46,47]. SMIT2 is highly expressed in mammalian kidneys and is responsible for the reabsorption of inositols into the bloodstream [48,49]. DCI uptake via SMIT2 is highly upregulated in the presence of insulin in human L6 myoblasts, which could explain the lower concentration of DCI in insulin-sensitive tissues and lower DCI re-uptake in the renal tubes in cases of hyperglycemia and insulin resistance [46,50,51], along with decreased epimerase activity [15,16]. The fact that inositol transport is highly dependent on glucose concentrations and insulin signaling limits its potential use in association with food (i.e., as an ingredient), where it would be necessary to adjust the timing, dose, and number of doses to the given metabolic profile of the user.

Inositols may also be transported by a described H^+^/myo-inositol transporter (HMIT) in a 1 H^+^ to 1 myo-inositol stoichiometric ratio [43,52]. Inositol HMIT transport is pH-dependent and phlorizin-sensitive [52]. HMIT is highly expressed in the brain, but its transcript is also detected in white, brown, and epididymal adipose tissues, and also in the kidney in rats [52]. HMIT is reported to transport MI, SI, DCI, and muco-inositol, but not allo-inositol, and is blocked by phloretin and phlorizin, which are well-described inhibitors of the Na^+^/glucose transporters SGLT1 and SGLT2 [52]. There are not many reports about the physiological role of HMIT in inositol transport in the peripheral tissues. HMIT differently controls inositol transport and signaling in the neurons and astrocytes along with SMIT1 and SMIT2, which may stand for a specific role of inositols in osmoregulation, insulin signaling, Ca^2+^ mobilization, and membrane composition in the brain [52,53,54,55]. A more detailed mechanism of inositol transport in the brain will be discussed in the other sections of this review.

MI, when available in the cell, is incorporated into PI. All MI-containing phospholipids are derived from PI, which is the most abundant form throughout the cell, constituting 10–15% of mammalian membrane phospholipids [56]. PI synthesis is performed next to the endoplasmic reticulum via PI synthase (PIS). This process requires cytidine diphosphate diacylglycerol (CDP-DAG) and MI. PIS has a low affinity for MI, hence MI availability is the rate-limiting factor for PI synthesis (ref). Some PI is channeled to the luminal face of the endoplasmic reticulum to later derive glycan PI or GPIs, which are acylated and transported to plasma membrane, serving as “anchors” for proteins in the external surface of the plasma membrane [57]. PIs may also be substrate for PI kinases, deriving PI-3, -4, and -5 monophosphate (PI(3)P, PI(4)P, and PI(5)P), which may suffer posterior phosphorylation and be converted into PI-3,5, -3,4, and -4,5 biphosphate (PI(3,5)P_2_, PI(3,4)P_2_, and PI(4,5)P_2_), the latter also rendering the triphosphate form PI(3,4,5)P_3_, as reviewed in [57,58]. Phospholipase C may cleave PI(4,5)P_2_ and form inositol (3,4,5) phosphate or IP_3_, which is subsequently derived into other IPs or recycled back to MI [58].

## 3. Insulin-Mimetic and Insulin-Sensitizing Properties of Inositols

### 3.1. Revisiting the Proposed Role of Inositols in Insulin Signaling

Two complementary mechanisms by which inositols modulate the insulin signaling pathway have been proposed [10,59,60] and revised [61,62]. Some questions regarding these models (depicted later) have been addressed, yet there are no clear answers. Recent findings may shed light on some of these unknowns and add other interactors in the signaling mechanism.

#### 3.1.1. Canonical Insulin Signaling

The classical mechanism of action in insulin signaling has been extensively described and reviewed [63,64,65]. Briefly, the binding of insulin to its receptor (IR) in target tissues promotes tyrosine autophosphorylation, recruiting IR substrates as the IRS and Shc proteins. Shc activates the Ras/MEK/ERK pathway, which accounts for mostly the growth-promoting effects of insulin. On the other hand, IRS1 and IRS2 continue the PI3K/Akt/mTOR pathway. IRS proteins recruit the p85 regulatory domain of phosphatidylinositol 3 kinase (PI3K), leading to phosphatidylinositol-3,4,5-triphosphate (PIP_3_), and activating the phosphorylation of Akt (also known as PKB). Full activation of Akt needs complementary phosphorylation by mammalian target of rapamycin (mTOR) complex 2 (mTORC2). Akt then mediates most of the insulin effects, as it phosphorylates and inhibits glycogen synthase kinase 3-β (GSK3-β), preventing the inhibition of glycogen synthase (GS) and leading to increased glycogen synthesis. Akt also promotes glucose uptake by the mobilization of glucose transporter 4 (GLUT4) and activates the mTOR complex 1 (mTORC1) via inhibition of tuberous sclerosis 1 (TSC1) and 2 (TSC1), leading to protein and lipid synthesis. Insulin signaling is a more complex process that involves major proteins that participate in the glucogenic pathway, such as fructose 2,6-bisphosphatase (FBPase-2), or the lipogenic pathway, like hormone-sensitive lipase (HSL), which are negatively regulated by protein kinase A (PKA) and also inactivated by Akt.

#### 3.1.2. Non-Canonical Insulin Signaling and the Role of IPGs

The role for inositols in insulin signaling has long been presumed, as early experiments showed the capacity of inositols to promote glycogen synthesis in the liver or as lipid synthesis in adipocytes. The paradigm of insulin signaling changed upon the discovery of insulin modulators that were produced upon phospholipase activity in GPIs, enhancing pyruvate dehydrogenase (PDH) activity and decreasing cAMP production [66,67]. Further research lead to the description of two types of IPGs based on their structure and activity. Type A IPGs (IPG-A) contain myo-inositol and d-glucosamine and inhibit cAMP production and AMPK activity, promoting lipogenesis. The others, named as type G IPGs (IPG-G), consist of a 3-o-methyl-d-chiro-inositol (d-pinitol) and galactosamine, promoting glycogenesis via mitochondrial PDH activation [68,69,70]. Larner et al. carried out isolation from beef livers and later confirmed the structure of an insulin second messenger (INS-2) with a molar ratio of 1:1 of 3-o-methyl-d-chiro-inositol (d-pinitol) and galactosamine linked by a β-1,4 bond [71] (Figure 1C). INS-2 that contains an inositol glycan structure of the so-called IPG-Ps is an allosteric modulator of PP2Cα [72], which is known to dephosphorylate and activate GS [73], PI3K [74], and inactivate AMPK [75]. INS-2 might also be present under the chelated form with Mn^2+^. Chelated INS-2 is an allosteric modulator of mitochondrial PDH phosphatase (PDHP) activity and promotes PDH-mediated glycogen synthesis [71]. It should be remarked that the structures of DCI-GPIs are still unknown and may not share structural similarity with MI-GPIs and differ in terms of the axial orientation of the phosphatidyl moiety, as reported by cleavage studies with synthetic DCI-GPIs (Figure 1C) [76].

Larner and colleagues described the role of a G_q/11_ protein as a putative pathway of insulin signaling [77], hence linking the activity of a phospholipase that could explain the release of IPGs from GPI and explain the crucial role for IPGs in insulin signaling. However, the exact structures of circulating inositols released by insulin stimulus are still unknown. This model proposed by Larner [10] raised some questions that were later added to new uncertainties in the review by Croze and Soulage [61]. Deep revision and the current data may help address some of these uncertainties.

The less widespread Müller’s theory [59,60] describes the role of IPGs in activating insulin signaling externally. This may be sustained by the observation that IPG internalization is not necessary to stimulate lipogenesis in rat epididymal adipocytes with a maximal activity of 47% of the maximum insulin response [78]. This theory is based on the existence of membrane detergent/carbonate-insoluble glycolipid-enriched raft microdomains (DIGs), which are formed by the high presence of cholesterol (hcDIG) or the low presence of cholesterol (lcDIG) in the plasma membrane. Some portion of insulin receptors seem to be associated with caveolins, mainly located in “caveolae”, which are structures in hcDIGs. GPI-anchored proteins would have a natural tendency to move to lcDIGs but are retained in hcDIGs by binding to a membrane protein, presumably p115. Insulin stimulus would lead to the activation of a GPI-PLC, which would release the IPGs. These IPGs may interfere in the binding of the GPI-anchored proteins to the receptor, allowing their displacement to the lcDIGs. This would also lead to a displacement of a protein kinase, pp59^Lyn^, previously attached to caveolin, which would mediate tyrosine phosphorylation on IRS1 or IRS2.

This theory would involve recognizing the existence of a GPI-PLC whose gene has not been identified in mammals, in addition to assuming that cholesterol microdomains are present in all cell types where insulin activity is shown, and this does not explain why IPGs may allosterically modulate intracellular elements of the insulin pathway. While this model cannot be ruled out, this may not represent a generalized mechanism and would serve as an additional route of complementary insulin signaling, but is not strictly necessary for insulin activity, rather than describing the main mechanism of action of the IPGs.

Whether there might be different IPGs contributing to insulin signaling depending on tissue or cell type might depend on the species and tissue proportion of inositol accumulation. Insulin markedly promotes the biosynthesis of DCI-GPIs after 15 min of addition to rat fibroblasts expressing the human IR, whereas a decrease in MI-GPI content is observed after 5 min of insulin treatment, which suggests that insulin promotes epimerase activity and conversion of MI to DCI [15]. DCI-containing IPGs might be the main mediators of insulin signaling, especially those involving glycogen synthesis. MI and SI are more prominent in the brain than DCI, whereas conversion of MI to DCI is far more prominent in fat, liver, muscle, or gonadal tissues [79]. It is foreseeable that DCI-IPGs would exert more important control over insulin signaling, effectively depending on the place of action.

Other debate has been raised between the intracellular or extracellular release of IPGs. The answer implies the interplay of three different proteins. Early experiments showed that IPGs are more likely to be extracellularly formed after GPI cleaving and are later actively transported in the cell. The presence of anti-IPG antibody blocks the activation of intracellular PDH, hence presuming that binding to extracellularly-generated IPGs to the antibody prevents access to the cell interior [80]. As such, some authors have described the existence of an ATP-dependent inositol glycan transporter that is stimulated upon insulin signaling. This plasmatic membrane transporter was first discovered in hepatocytes and has been well described [81]. Thus, it is a putative IPG transporter that would support the extracellular release of IPGs.

Since IPGs are part of the polar head of GPIs, their release relies on phospholipase activity. The proposed mechanism implies an alternate pathway to tyrosine phosphorylation or IRs, with IRs also coupling to a heteromeric protein G_q_ and the activation of a GPI-phospholipase [10]. Both GPI-PLC and GPI-PLD have been proposed as candidates. Early experiments have determined the generation of IPGs under the activity of bacterial GPI-PLC and GPI-PLD [60,68,71,82]. However, gene encoding for a mammalian GPI-PLC has not been identified yet. Presumably, insulin mediates the generation of IPGs through a GPI-PLD, as has been described [83,84]. GPI-PLD expression is ubiquitous throughout all tissues and is especially prevalent in the liver and circulating in plasma [85]. Current studies appoint the relevance of GPI-PLD in insulin resistance. Significantly increased levels of GPI-PLD have been identified as a novel biomarker of early prediabetes in humans [86] and early stages of latent autoimmune diabetes in adults and those with type 2 diabetes [87]. It has been observed that both insulin and glucose stimulate the secretion of GPI-PLD in rat pancreatic islets [88]. GPI-PLD levels also seem to be higher in the pancreas under islet hyperactivity and lower in the liver from insulin-resistant (*ob/ob*) mice [88].

In relation to the above, despite the fact that there is no evidence of the mammalian gene for GPI-PLC in humans, this possibility cannot be ruled out yet due to identification of a GPI-PLC-like protein in bovine brains and rat intestines [89,90], but also to the lack of knowledge of the chemical structure of the various IPGs that can be generated in the body. These doubts are raised by the experiments carried out with synthetic IPGs. In one study, it was observed that the phosphate group that binds carbon 1 of inositol to the membrane lipid needs to be maintained after cleavage of phospholipase and forms a cyclic linkage with carbons 1 and 2 of inositol for certain synthetic IPGs that have an insulin-mimetic activity [70]. This is only achieved through the action of a PI-PLC, since the hydrolysis of the phosphate is carried out on the O^−^ radical bound to the membrane lipid, whereas a GPI-PLC performs a cleavage on the O^−^ radical of inositol, maintaining the phosphate group in the lipid after the release of the IPG (Figure 1C) [76]. In addition to this, synthetic DCI-GPI anchors with α(1→2) linkage of glucosamine and DCI cannot be a substrate for PI-PLC hydrolysis, but this can be mediated by GPI-PLD (Figure 1C) [76]. This suggests that possible DCI-GPIs are structurally similar to MI-GPI anchors with α(1→6) linkage of glucosamine and that MI relies on GPI-PLD activity. Fagopyritols are galactose and DCI analogs found in plants and are classified according to the binding (type A with galactose-α(1→3)-DCI linkage and type B with galactose-α(1→2)-DCI linkage). Fagopyritol B1, a galactose-α(1→2)-DCI, is a structural analog of the core of the proposed DCI-GPI anchors and has a more powerful insulin-mimetic effect than free DCI, highlighting the possible role of DCI-IPGs in insulin signaling (Figure 1C) [91].

PI-PLC cannot hydrolyze and release IPGs with cyclic phosphate when inositol carbon 2 is palmitoyl-acylated, which is often the case for non-anchored protein-free GPIs [92]. In contrast, GPI-PLD may be cleaved when inositol groups are acylated, supposedly releasing acylated IPGs (A-IPGs) (Figure 1C) [92]. Synthetic A-IPGs also show a strong insulin-mimetic activity [93]. Non-protein linked GPIs are intermediate GPIs, as they quickly bind proteins when reaching the plasma membrane surface. However, non-protein-linked GPIs have been observed to reside both in the inner and outer leaflets of the plasma membrane [94]. GPI-PLD release of acylated IPGs (A-IPGs) is speculated to occur in the intracellular compartment, yet this has not been corroborated [94]. The fact that anti-IPGs block some of the insulin-mediated actions suggests that intracellularly-released A-IPGs have a minor, yet complementary, role in inulin signaling [80]. Besides insulin, GPI-PLD expression is associated with lipid levels [95,96,97] and its activity is also associated with triglyceride [98] and lipid metabolism in the liver [99], which may be somehow related to the improved lipid profiles of patients suffering from metabolic diseases after supplementation with inositols like DCI [100].

Assuming this approach, inositol supplementation may restore pathologically low levels of IPGs, given that the rate-limiting aspect of GPI synthesis is cytosolic-free inositol supply, as phosphoinositol synthase has a relatively low affinity [57]. Thus, a higher concentration of inositols, especially DCI, which is much more scarce than MI and has a more prominent role in insulin signaling, supports the idea that inositol supplementation would help the synthesis of DCI-GPIs and later form DCI-IPGs when insulin epimerase’s activity is diminished in insulin resistance.

Albeit that this model seems to be a fairly close approach to the true role for inositols as insulin-sensitizers (Figure 2), data on supplementation with inositol derivatives might question whether IPGs are the only way for inositols to modulate insulin signaling.

### 3.2. Are Inositols Direct Insulin Mimetics Rather than Insulin Sensitizers?

Apart from the proposed model of IPGs in insulin signaling, some early in vitro studies of inositol supplementation have shown the direct effect of inositol isomers in insulin signaling apart from IPG activity. It was shown that a 1 mM dose of DCI, DPIN, l-*chiro*-inositol (LCI; *cis*-1,2,4-*trans*-3,5,6-ciclohexanehexol), EI, and muco-I stimulated glucose uptake in rat L6 myotubes in vitro and also promoted that translocation of GLUT4 to the plasma membrane in L6 myotubes in vitro and the skeletal muscles of rats ex vivo to a similar extent as 100 nM of insulin [101]. It should be noted that cells were grown in a medium supplemented with fetal bovine serum (FBS) and starved 18 h prior to the glucose uptake assay with a medium containing 0.2% FBS, so the possibility of insulin traces remaining in the cell culture media should not be ruled out, as these could account for the inositol effect on GLUT4 translocation and glucose uptake at the given concentrations. MI was also supplemented at same concentrations and did not elicit any insulin-mediated response on glucose uptake [101].

When administered to endothelial cells in vitro, both 1 mM of MI and DCI promoted an increased phosphorylation of Akt, ERK1, and ERK2 in human vascular endothelial cells (HUVEC) to a greater extent than 100 nM of insulin [102]. Since cells were serum-starved before MI and DCI treatment, MI and DCI induced kinase phosphorylation in the absence of insulin. It is noteworthy that the Ras/MEK/ERK signaling pathway was also involved, as inositols are regarded to exert insulin-sensitizing effects on the PI3K/Akt/mTOR signaling pathway.

Our recent study has shown that the administration of DPIN to fasting rats promotes a significant reduction of circulating insulin without affecting plasma glucose levels [103]. These results may imply a direct action of DPIN on insulin signaling. An increase in ghrelin levels was also observed upon DPIN administration, which could account for the decreased secretion of insulin in pancreatic β cells [103].

These results show that inositols may act on upstream regulators of insulin signaling. Some experiments agree with this hypothesis. Sequoyitol pretreatment enhances insulin signaling with increased phosphorylation of IRS1 and Akt in HepG2 hepatocytes and 3T3-L1 adipocytes [104]. Interestingly, sequoyitol pretreatment also reverses decreased IR autophosphorylation in the presence of tumor necrosis factor (TNF-α), a well described inhibitor of IR activity [104,105]. TNF-α is known to also inhibit SMIT expression in cultured endothelial cells [106]. Inositol depletion might partially explain the insulin-sensitizing effect of sequoyitol on insulin-resistant cells but does not account for enhanced IR autophosphorylation.

Increased IR autophosphorylation has also been observed in primary hippocampal neurons from rats when administered 100 μM of DCI, DPIN, or INS-2 in a similar way to 1 μM insulin treatment [107]. Moreover, after media replacement with a serum-free HEPES buffer, DCI administration has been shown to promote IR internalization, a mechanism required for ERK activation, in a similar way to insulin [107]. Given that insulin was depleted from the media, again, this mechanism of IR trafficking from dendrites to soma elicits a direct effect of DCI as a free inositol in insulin signaling.

The way free inositols participate in the insulin signaling pathway remains unknown. Given that PLD levels are relatively high in all tissues, insulin stimulation of GPI-PLD might not be crucial for its activity. Somehow, a stimulus would be needed to increase IPG production, which would rely on the basal activity of GPI-PLD [108]. Based on the results obtained both in vitro and in vivo, the group of Ashida suggested an insulin-independent mechanism, implying the activation of PI3K and/or AMPK [101,109]. It has been described that INS-2 allosterically modulates PP2Cα [72], which is known to dephosphorylate and activate PI3K [74] and inactivate AMPK [75]. IPG production could account for GLUT4 translocation and enhanced glucose uptake in muscles. However, inositol supply and increased IPG production does not account for the direct actions of inositols regarding IR autophosphorylation, since their target starts insulin signaling downstream of PI3K and possibly IRS1.

Although unlikely, some hypotheses have arisen regarding the means of IR activity. One of them is the possible allosteric modulation of IR by inositols. A study has identified the molecular docking of MI with active sites of PPARγ, GLUT4, and IR [108]. However, contrasting docking analysis requires specific studies of protein–molecule inhibition or inhibition of the protein target to determine the correlation of effective interactions in a biological system and in silico predictions. Moreover, the fact that MI is predicted to interact with different molecules of the same signaling pathway undermines the reliability of this mechanism. Another possibility could involve the repression of negative regulators of IR. Protein tyrosine phosphatase 1B (PTP1B) is a well-described inhibitor of IR tyrosine kinase activity. PTP1B is activated under different stimuli, including the presence of TNF-α [110]. Other known inhibitors of IR are c-Jun N-terminal protein kinase (JNK) or suppressor of cytokine signaling 3 (SOCS3). It is still unknown if inositols may interact and downregulate one of the inhibitory pathways of IR autophosphorylation,

Mechanisms of the insulin signaling pathway may involve different downstream elements, but they all share a common activation of IRS1/2, PI3K, and Akt. In order to elucidate the target for free inositols, we propose an in vitro study of cultured hepatocytes or myocytes, as highly-responsive cells to insulin stimulus, and the blockade, one by one, of elements composing the cascade of the insulin signaling pathway (IR, IRS1, PI3K, PDK, Akt, mTOR, and AMPK) in a top-down manner. This could be easily achieved via the use of small interfering RNA (siRNA) and transient siRNA-mediated knockdown of IR and their downstream elements [111,112]. Measuring the GLUT4-translocation response of insulin-depleted cells to inositol addition could determine the exact point at which inositols enhance IR signaling in the absence of insulin. In vitro studies let us control medium conditions and eliminate external elements that could interfere with the insulin sensitivity. Alternatively, the use of specific inhibitors for each element could be considered. Identification of inositol targets would make way for further analysis of the exact mechanism of interaction, as well cyclitols, regarding their specificity.

### 3.3. Putative Role of Inositols in IGF-1 Signaling

Insulin and insulin-like growth factor 1 (IGF-1) are both hormones with a high structural similarity and share some cross-reactivity due to the low-affinity binding of insulin to the IGF-1 receptor (IGF-1R) and from IGF-1 to IR. The existence of active IGF-1/IR heterodimers has also been demonstrated, although their physiological role has not been fully described. In contrast to insulin, IGF-1 is released in the liver and is stimulated by the growth hormone and its function is strongly anabolic. IGF-1 circulates as a ternary complex consisting of IGF-1, IGF binding protein 3 (IGFBP-3) or 5 (IGFBP-5), and the acid labile subunit (ALS), avoiding IGF-1 non-specific insulin-like hypoglycemic activity. The metalloproteinase pregnancy-associated plasma protein A2 (PAPP-A2) is involved in the proteolysis of the IGF-1 ternary complex, releasing free and active IGF-1 on target tissues [113].

Like insulin, IGF-1 is also able to stimulate GPI cleavage and IPG formation, as seen in vitro in 3T3 fibroblasts, BC3H-1 myocytes, and Chinese hamster ovary (CHO) cell lines [114,115,116]. Moreover, antibody binding to IPGs formed after the addition of IGF-1 blocks the growth-promoting effect on the ears of chicken embryos [117]. However, the addition IPGs without the presence of IGF-1 has a negligible effect on growth, which suggests that IPG formation is necessary, but not necessarily able to promote an IGF-1-mediated growth effect [117]. This effect is likely mediated by IPG-A, since IPG activity has been measured by its capacity of inhibiting PKA [117]. Another study showed that the addition of antibodies against IPG-P blocked the stimulatory effects of both IGF-1 on progesterone synthesis by swine ovary granulosa cells [118]. However, in adult rat hepatocytes, insulin mediates GPI cleavage and IPG formation, and it has been observed that fetal hepatocyte formation of IPGs is dependent on IGF-1 but not insulin activity. Furthermore, the addition of isolated IPG-P, but not insulin, has reduced the activity of glycogen phosphorylase (the rate limiting enzyme for glycogen hydrolysis) [119].

The results mentioned earlier suggest a role for IPGs as putative mediators of IGF-1 signaling. The involvement of IPGs on IGF-1 activity seems to be complementary to the canonical IGF-1 activation of the PI3K/Akt/mTOR and Ras/MEK/ERK pathways during development, acquiring a more prominent role for insulin signaling in adulthood. Despite these results, the study of inositols for IGF-1-like properties has long been neglected and no more recent data are available, including a lack of complementary in vivo results. It is yet to be unveiled whether inositol deficiencies may cause growth and development problems due to poor IGF-1 signaling. The addition of free inositols as compared to inositol glycans during postnatal growth may provide deeper insight inositol mechanisms of action for insulin/IGF-1 signaling.

## 4. The Antioxidant Capacity of Inositols

In addition to the modulation of insulin signaling, inositols are polyols that might act as modulators of oxidative metabolism, helping to decrease the burden of oxidative stress. Oxidative stress is the most common factor responsible for metabolic disturbances caused by insulin resistance. Under normal reduction-oxidation conditions, physiological metabolism, especially via aerobic processes, produces a series of sub-products called reactive oxygen species (ROS) that include superoxides (O_2_^−^), hydrogen peroxides (H_2_O_2_), and hydroxyl radicals (OH^−^) as part of the oxidation of metabolites. Normally, antioxidants present in organisms can compensate the generation of ROS, as they accept electrons of negatively charged oxygen molecules, deriving them into H_2_O.

A common feature of insulin resistance is the elevated production of cytokines such as TNF-α and interleukin-6. A pro-inflammatory state contributes, along with hyperglycemia and decreased insulin signaling, to a deregulated metabolism and excessive generation of ROS. Oxidative stress in adipose tissue, along with an exacerbated release of cytokines, also promotes a pro-inflammatory state that contributes to the development of insulin resistance, diabetes, and concomitantly increases the risk of obesity-associated metabolic syndrome. In presence of nitric oxide (NO), a quick cross-reaction with O_2_^−^ produces cytotoxic peroxynitrite (ONOO^−^) as part of the reactive nitrogen species (RNS). Both ROS and RNS attack biological components of cells, including DNA, RNA, protein, or lipid peroxidation, causing severe damage to plasma and organelle membranes [120]. Oxidative/nitrosative stress during insulin resistance causes endothelial dysfunction and vascular complications or atherosclerosis. Endothelial cell production of NO is promoted by insulin via PI3K/Akt signaling, which leads to the activation of endothelial nitric oxide synthase (eNOS) [121,122]. However, decreased insulin signaling, along with elevated NADPH oxidase activity and increased generation of O_2_^−^, leads to a low bioavailability and bioactivity of NO [123].

In phenolic compounds, hydroxyl groups can transfer their hydrogen to negatively charged free radicals (R^−^) in order to be stabilized as neutrally charged radicals (RH). As inositols are polyalcohol molecules, it has long been presumed that they possess antioxidant potential due to the presence of hydroxyl groups. The first approaches to inositol derivative molecules focused on the antioxidant potential of the inositol phosphorylated derivative phytic acid (IP_6_) as an iron (Fe^3+^) chelator. In normal conditions, free radicals are generated in the Fenton reaction by the oxidation of Fe^2+^ as follows: Fe^2+^ + H_2_O_2_ → Fe^3+^ + HO• + OH^−^. However, the sequestering of Fe^3+^ by IP_6_ leads to a rapid depletion of Fe^2+^, which is oxidized by molecular oxygen (O_2_), but not by H_2_O_2_, hence blocking free radical formation [124,125]. Antioxidant activity of IP_6_ is seen to be especially relevant for the xanthine/xanthine oxidase system, which generates H_2_O_2_ by consecutive hypoxanthine to xanthine and xanthine to uric acid oxidation reactions [126].

The beneficial effects of free inositols in oxidative stress have also been related to radical scavenging properties. An in vitro study showed that the addition of DPIN exhibited dose-dependent inhibition of superoxide and nitric oxide formation [127]. In endothelial cell cultures under high glucose conditions, the addition of DPIN, DCI, and synthetic 3,4-dibutyryl-DCI (db-DCI) has been shown to dose-dependently scavenge superoxide in an xanthine/xanthine oxidase system [128]. It was shown that db-DCI was most effective at reducing ROS levels and exhibited an Fe^3+^-related mechanism of action, suggesting that db-DCI acts similarly to IP_6_, although the detailed mechanism of this has not been determined yet [128]. Interestingly, quebrachitol has also been described as an active component displaying a ONOO^−^ scavenging activity [36].

The efficacy of inositols as antioxidants may also be attributed to an enhanced activity of antioxidant enzymes. Recent studies on Jian carp have shown that MI supplementation increased the activities of catalase (CAT), glutathione peroxidase (GPx), and glutathione reductase (GR) in copper (Cu)-induced toxicity, but also superoxide dismutase (SOD) and glutathione-S-transferase (GST), both in normal and Cu-induced damage conditions [129,130]. DPIN has also showed an enhancement of endogenous antioxidant activity. DPIN at a dose of 200 mg/kg inhibits oxidative stress caused by 7,12-dimethylbenz(a)anthracene (DMBA) in rats, along with an increased activity of the antioxidant enzymes SOD, CAT, GPx, and GST [127]. In a mouse model of cisplatin-induced oxidative stress, the administration of DPIN increased GSH, SOD, and CAT activities [131]. The administration of an DCI-enriched extract to streptozotocin-induced diabetic mice significantly increased glutathione (GSH) and decreased malondialdehyde (MDA) in the liver, accompanied by decreased pro-inflammatory TNF-α and increased anti-inflammatory IL-6 and interferon gamma (IFN-γ) in the sera [132].

As described previously, endothelial dysfunction is a common pathology derived from insulin resistance. Regarding this, use of inositols yields a synergistic effect for both antioxidant and insulin-sensitizing activities. The administration of MI and DCI in HUVEC cells promotes Akt phosphorylation [106]. The in vitro addition of DPIN, DCI, and db-DCI impaired contraction by the eNOS inhibitor L-NAME and increased NO effectiveness [128]. It has also been shown that db-DCI decreases reduced PKC activation, hexosamine pathway activity, and advanced glycation end products to basal levels in high glucose conditions [128].

Given the increasing problems arising from an unhealthy diet and living conditions, dietary use of inositols should be considered because of their antioxidant and insulin-sensitizing properties (Figure 3). It is still necessary to determine the exact mechanism of free radical scavenging in inositols. Structure differences are likely to contribute to the net antioxidant capacity, as differences have been observed between the sugar alcohols MI and DCI and methyl derivatives DPIN or db-DCI. There are no available research data on the antioxidant activity of IPGs, since PI-derivative structures like IP_6_ could be active compounds of inositol activity.

## 5. Inositols in the Brain

### 5.1. Sources and Distribution of Inositols in the Brain

For many years, inositol disposition in the brain has gained much attention due to the observation that inositol levels are 7-fold higher in the cerebrospinal fluid relative to plasma, and some 50- to 200-fold higher in the brain, in addition to several reports of altered MI and SI levels with different neuropathologies [133]. As in the rest of the body, MI is the main inositol present in mammalian brain tissue, followed by SI and small traces of DCI, NI, EI, or muco-I [12,13,25].

Inositol supply in the brain comes from three major sources, namely, the recycling of PI derivatives, de novo synthesis, and inositol active transport from the peripheral tissues. The synthesis of inositols produced in the brain occurs to a lesser extent relative to the peripheral tissues [37]. The activity of MIPS has been detected in the microvasculature of mammalian brains [134]. In vitro studies with neuroblastoma cells have shown that the expression of the inositol synthesis enzyme IMPase is necessary for GSK3-α but not GSK3-β activity [135]. However, other previous results did not detect IMPase activity in vitro in NT2-N neurons [136]. There are still some discrepancies regarding whether de novo synthesis and inositol recycling are major sources of free inositol for normal neuronal and glial activities without the need for active transport from peripheral sources of inositols. Homozygous SMIT1 KO animals show remarkedly decreased MI levels in the whole brain, especially in the frontal cortex (55% reduction) and hippocampus (60% reduction), but normal levels of PI, IP_5_, and IP_6_, which may suggest de novo MI synthesis maintains PI-derivative levels in the brain [137,138,139]. However, IMPase KO mice show a 65% decrease in IMPase activity but normal MI levels in the hippocampus [140]. We suspect these differences may rely on compensatory mechanisms of inositol replenishment, a pool of inositol reserves as PI, or differences in inositol content and metabolism in neurons versus glia. Although PI intracellular levels do not change in SMIT KO mice, overexpression of SMIT in transfected cells has shown the same PI levels as the control cells and intracellular PIP and PIP_2_ levels increased, which may suggest a different or minor pool for PI-derived signaling molecules responsive to SMIT or MI levels and these are not sensitive enough to contribute to the total PI pool in the cell [141,142].

The distribution of MI in the brain is unequal and may be representative of particularities in regional activity. MI levels are higher in hypothalamus relative to the hippocampus, as detected by ^1^H-magnetic resonance spectroscopy (MRS) in mouse brains [143]. MI uptake is also produced at a higher rate when compared to the hippocampus, cortex, caudate, or cerebellum [144] in rat brains. The hypothalamus is adjacent to the third ventricle, where the blood–brain barrier permeabilizes and provides access to metabolic signals from the peripheral tissues like insulin, glucagon, leptin, gherlin, or glucose itself [145]. Variation in regional MI might be due to different expressions of the inositol transporters SMIT1 and SMIT2. Cerebellar mRNA expression of SMIT1 and SMIT2 is higher than hippocampal and cortical expression in mice [47]. Apart from SMIT1 and SMIT2, HMIT has gained relevance as it is expressed predominantly in the brain, especially in the neuronal population of the human hippocampus and cortex, as determined by immunocytochemistry [52,54]. Analysis of RNA expression of HMIT in rat brains has shown that the HMIT transcript is expressed predominantly in the brain, with higher expression found in the cerebral cortex, hippocampus, hypothalamus, cerebellum, and brainstem [52]. Inositol HMIT transport is pH-dependent and phlorizin-sensitive [52].

Importantly, different presences of SMIT1, SMIT2, and HMIT have been detected in astrocytes and neurons. A study in cultured astrocytes and neurons showed that HMIT and SMIT1 are more present in astroglia than SMIT2 and may contribute to a higher uptake of MI due to their affinity [55]. On the other hand, SMIT1, SMIT2, and HMIT are all expressed in neuronal cells, where SMIT2 is expressed at higher levels [55]. Even though HMIT has been suggested to be relocated actively between plasma membrane and vesicles via exocytosis in regions of nerve growth, further studies have shown that HMIT is not actively expressed in the cell membrane of human neurons and does not participate in inositol internalization [53,54]. HMIT is co-stained with Golgi markers in neurons, indicating that it could participate in vesicular inositol trafficking. Since IP_3_ is a substrate for HMIT transport, it has been speculated that the role of HMIT would be more committed to the regulation of intracellular IP_3_ levels and Ca^2+^ signaling instead of participating in inositol internalization in neurons [54]. The expression of HMIT has also been detected in astrocytes and it seems to be localized both in intracellular and plasma membrane, as depicted by immunochemistry [52]. HMIT shows high capacity/low affinity transport kinetics and is relevant for MI transport under physiologically relevant MI concentrations, whereas under intracellular acidic conditions or lower extracellular MI conditions, SMIT1 and SMIT2 (to a lesser extent) are the main mediators of inositol uptake in primary cultures of mouse astrocytes [55]. This suggests that inositol transport in neurons and astrocytes is regionalized and mediated by different transport systems, which could be associated with a specific role of inositols in the intracellular signaling mechanism.

When incorporated into phospholipids, PI derivatives show specific functions in the nervous system, as reviewed [146]. Briefly, PI(3)P is important for the hippocampal regulation of GABAergic inhibitory transmission, PI(5)P regulates Notch cell signaling, PI(4,5)P_2_ is involved in different processes of neuronal excitability, PI(3,5)P_2_ affect glutamatergic signaling, and both PI(3,4)P_2_ and PI(3,4,5)P_3_ have a role in dendrite development.

The recent interest of the inositol derivative lysophosphatidylinositol (LPI) as a central regulator of memory and inflammatory processes should also be highlighted. LPI is formed by the action of phospholipases A1 (PLA1) and A2 (PLA2) on PI and serves as an intermediate for the synthesis of endocannabinoid 2-arachidonoylglycerol (2-AG). However, LPI has an important role in controlling neuronal excitability and responsivity to external stimuli, as it acts as a putative ligand for cannabinoid G protein-coupled receptor 55 (GPR55). In the periphery, GPR55 is known to modulate and increase insulin secretion in beta-pancreatic islets via a mechanism involving the mobilization of intracellular Ca^2+^ [147]. GPR55 is also known to be involved in energy metabolism and pain sensation [148,149]. Specifically in the brain, GPR55 has been shown to be expressed in the hippocampi of mice and rats and is localized in the CA1 and CA3 layers of pyramidal cells [150]. The application of LPI to hippocampal mouse slices enhances the long-term potentiation of CA1 neurons [150]. Moreover, central administration of LPI and GPR55 agonists promotes procedural memory and provokes changes in spatial memory [151,152]. Central actions of GPR55 seem to rely on its ability to modulate intracellular Ca^2+^ presynaptically and boost neurotransmitter release, as observed in the hippocampal CA1 to CA3 subregions of mice brain slices [153]. These results suggest that the inositol derivative LPI is able to regulate cognitive processes through the activation of GPR55. A summary of inositol distribution in the brain and the main activities can be found in Table 2.

A more detailed description of the role of inositols in the brain could clarify the possibilities of their use as nutraceutical treatments. Nevertheless, when considering the external supply of inositols in the brain, it should be considered that the administration of inositol derivatives like SI decreases the concentration of MI, which may represent a shift in the inositol equilibrium, promoting MI degradation to stabilize brain homeostasis [18]. Hence, the administration of inositol derivatives should be tightly regulated in order to avoid an imbalance in inositol homeostasis.

### 5.2. Inositols as Osmolytes in Astrocytes

Inositols are described as osmolytes in plants, protecting them from heat, high salinity, and drought stress [33]. In mammals, higher concentrations of inositols in the brain, with respect to the peripheral tissues, stand mainly for their osmoregulatory role in astrocytes, as multinuclear NMR studies have suggested [166]. SMIT gene expression is increased under osmotic stress conditions and rapidly decreased when iso-osmolarity is reestablished [167]. This mechanism allows astrocytes to adapt their size in order to reduce the impact of an ionic imbalance in extracellular media. In hypotonic conditions, the volume-sensitive organic osmolyte anion channel (VSOAC) mediates the rapid efflux of inositol along with other osmolytes, leading to cell shrinkage [168].

This property of MI fluctuation in glial cells has led to its correlation as a widely accepted marker for astrogliosis. An ^1^H-MRS study in brains showed reduced MI levels after a traumatic brain injury, which was later normalized over time [169]. This decrease in brain MI may be a result of astrocyte cell death or either a mechanism of osmoregulation to prevent the development of brain swelling [170]. However, the activation of microglia and astrocytes leads to the accumulation of osmolytes and increased cell size, reflecting a higher MI content [171]. MI accumulation in astrogliosis is also a common feature observed during aging in the hippocampus and cortex [172], which seems to reflect changes in astroglia cell metabolism, a chronic inflammatory state, and oxidative stress [173,174].

As observed during aging, elevated MI levels in the brain are also observed during previous stages of mild cognitive impairment (MCI), AD, and pre-AD patients with Down’s syndrome (DS), with a negative correlation between MI levels and cognitive performance [154,155,156,157]. Altered MI levels are also observed with bipolar disorder [175]. Changes observed in MI levels localized in the hippocampus and frontal cortex, involved several functions, including memory and task decision, make MI a valuable biomarker of early stages of cognitive decline as it can be detected in vivo in early stages of cognitive decline and correlates time-dependently with its development.

### 5.3. Changes in Inositol Derivatives and Excitability in Neurons

It is well established that inositols are osmoregulators in astrocytes. Since this may also be true for neuronal populations, an important role of SMIT transporters and inositol transfer within the cell has been found as intrinsic modulators of neuronal excitability.

Membrane potential is highly influenced by the efflux of K^+^ through ion-, voltage-, or metabolic-dependent channels, which are responsible for the inhibition of cell excitability. As a substrate, the inositol derivative PIP_2_ acts as a “metabolic sensor” and interacts with the Kir6.2 and SUR1 subunits of ATP-sensitive K^+^ channels (K_ATP_), stabilizing the open state of the channel (as opposed to ATP, which stabilizes the closed state of the channel), which is an important mechanism of the pancreatic release of insulin [176,177,178,179]. In neurons, excitability is tightly controlled by G protein-gated inwardly rectifying potassium (GIRK) channels and muscarine-sensitive voltage-gated potassium (Kv) channels, where KCNQ2 and KCNQ3, forming the heteromer KCNQ2/3, are major contributors in the hippocampus for M-current, a non-inactivating K^+^ that defines phasic versus tonic firing [159,160]. As an intracellular signaling molecule, PIP_2_ promotes the opening of GIRK [180] and KCNQ2/3 [181,182]. According to this, the KCNQ2/3 current is inhibited when PIP_2_ is depleted via the activation of Gq-coupled receptors and the subsequent PLC cleavage of PIP_2_ [181].

Regarding the later information, PIP2 is necessary for the modulation of KCNQ2/3 heteromer activity, but recent data point to a SMIT-mediated control of channel function. SMIT1 and SMIT2 interact physically and colocalize with KCNQ2/3 in sciatic nerve nodes of Ranvier and in axon initial segments [158,183]. In the absence of MI, SMIT1 co-expression with KCNQ2/3 modulates channel ion selectivity, gating, and pharmacology, making KCNQ2/3 less sensitive to extracellular K+ and promoting M-current [183]. However, when MI is added, KCNQ2/3 currents augment with a seemingly increased PIP_2_ synthesis [158]. This fact is supported by increased PIP and PIP2 levels in hypertonic medium and faster cell recovery after drug-mediated suppression of KCNQ2/3 current [141]. Thus, both SMIT and MI synergistically promote hyperpolarizing currents mediated by KCNQ2/3. Interestingly, GABA modulates KCNQ2/3 in a similar way as SMIT1. The activation of KCNQ2/3 seems to be mediated by SMIT1 through KCNQ2, whereas GABA communicates with KCNQ3. GABA binds to a S5 residue in both KCNQ2 and KCNQ3, mediating a conformational change that leads to loss of SMIT1 influence over KCNQ2/3. Since SMIT1 also binds the same S5 residues, its co-expression reduces GABA potentiation with respect to KCNQ2/3. Thereby, the presence of GABA decreases SMIT1 influence over KCNQ2/3 and vice versa as a negative feedback mechanism [160].

The finding that inositol regulates neuronal excitability is especially relevant regarding correlations of altered KCNQ conductance in several models of neurodegenerative diseases. Alzheimer’s disease (AD) has been correlated to a reduced expression of the GirK2, GirK3, GirK4, KCNQ2, and KCNQ3 subunits and genes encoding for the antioxidants SOD, 8-oxoguanine DNA glycosylase (OGG1), and monoamine oxidase A (MAO-A) in rats [184]. Familial-inherited epilepsy has also been associated with mutations in KCNQ2 and KCNQ3 [185,186,187]. The expression and function of KCNQ2 [188] and KCNQ3 [189] are both also altered in cases of bipolar disorder. Crosstalk between SMIT/MI and KCNQ channels is an important issue, as changing intracellular levels of MI and thus PIP_2_ levels may modulate neuronal excitability in the hippocampus. The fact that many neuropathologies are related to either SMIT- or KCNQ2/3-altered function highlights the importance of inositol in normal neuronal function.

## 6. Neurodegenerative Diseases: Perspectives for the Use of Inositols

### 6.1. Alzheimer’s Disease

#### 6.1.1. Inositols and the Amyloid Pathology

AD dementia is the main clinical entity contributing to the increased prevalence of dementia. In the absence of effective therapies to avoid the deposition of β-amyloid (Aβ) plaques, the main pathogenic factor studied in AD, efforts are also focusing on procedures to delay the cognitive decline associated with this anomalous protein accumulation.

One of the most characteristic features of AD development is the aberrant production and deposition of Aβ peptides, either in Aβ_40_ or Aβ_42_ fragments. Amyloid precursor protein is subject to protease activity by β-secretase (BACE1) and γ-secretase. Presenilin-1, the catalytic subunit of γ-secretase, is regarded as a major contributor of increased Aβ production. Notably, γ-secretase also contributes to the cleavage of other substrates like Notch, which is important for cell differentiation in embryogenesis and adulthood [190]. The accumulation of Aβ oligomers extracellularly leads to the appearance of Aβ-derived diffusible ligands (ADDLs), which are highly neurotoxic and mediate the disruption of neuronal synapses, the depression of signaling, tau hyperphosphorylation (another well studied factor in AD), the disruption of normal autophagic processes, the generation of ROS and RNS, and cell death [107]. The combination of the above factors is seen to be the cause for cognitive impairment and memory loss.

The tendency of Aβ accumulation seems to be facilitated by binding to peroxidized PI lipids as a consequence of oxidative stress, inducing the conformational secondary structure of the β-sheet [191]. Based on this, it was hypothesized that free myo-inositol and inositol stereoisomers might interfere with Aβ aggregation and fibril formation competing for Aβ-PI lipid binding and stabilizing soluble forms of Aβ [23,24]. Early studies in vitro proved that MI, SI, and EI induced formations of stable β-structures of Aβ_42_, but these structures did not result in Aβ_42_ progression to fibrillar structures at a 1:1 ratio (by weight). The same studies showed that non-fibrillar Aβ_42_ oligomers, which may cause neuronal toxicity as well, were non-toxic in the presence of EI and SI at a ratio of 1:20 [24]. DCI, however, was unable to avoid Aβ_42_ fibrillar conformation and toxicity, which is suggested to be due to the necessity of the hydroxyl groups to be oriented at positions 1, 3, and 5 or alternatively 2, 4, and 6 in order to stabilize the inositol-Aβ_42_ complex structure [24].

Further studies in vivo have shown that the prophylactic administration of SI and EI is effective for preventing cognitive decline and Aβ aggregation at early stages in a mouse model of AD (TgCRND8) and 1-month treatment of SI, but EI could not also reverse disease development [192]. Positive effects with SI were corroborated with decreased astrogliosis and an improvement of synaptic transmission, observed by increased levels of synaptophysin [192]. A later study corroborated higher levels of SI in the brain after ad libitum administration to TgCRND8 mice, with reduced Aβ_40_ and Aβ_42_ brain levels and decreased plaque formation [18]. Moreover, Aβ cerebroventricular-injected rats with ad libitum access to SI in drinking water had improved performance for learning tasks than non-SI treated rats [193]. Since SI showed better results than EI and other inositol stereoisomers, later studies focused on the SI effect in the same and other murine models of AD, showing improvements in damaged cortical microvasculature caused by Aβ accumulation and improved spatial learning [194,195,196].

These studies have been substantially reviewed and have raised the not-so-feasible properties of SI (and, to a lesser extent, EI) and its ability to decrease Aβ aggregation and neuronal damage. Some complaints have arisen from the concentration of SI or EI required for interaction with Aβ and the disruption of fibril formation [197]. Other studies have shown the inefficacy of SI at a ratio of 1:10 for inhibiting Aβ aggregation and toxicity, as well as cell death for PC-12 cells and mixed primary rat hippocampal neurons mixed with glial cells [198]. Others have reported that SI binds weakly to other minor-contributing amyloid fragments like Aβ_25−35_ (formed in aging brains after Aβ_40_ cleavage) [199,200]. Despite these concerns, SI levels are dramatically increased in the brain after ad libitum access, thus being able to interfere in Aβ toxicity, as observed in animal models [18].

These results led to a phase II clinical study of SI (named as ELND005) for mild-to-moderate AD. The administration of SI at a 250 mg dose resulted, after 78 weeks of treatment, in a slight improvement in neurological performance, increased brain ventricle volume, and lower levels of CFS Aβ_42_, accompanied by higher levels of SI in CSF, as observed in animal models [201]. Further clinical trials have focused on SI effects on agitation and aggression in AD. Despite these results, higher doses of SI resulted in early deaths (4 deceased for 1000 mg SI dose and 5 deceased for 2000 mg SI dose) [201]. SI toxicity may have been caused by renal failure, as uric acid was decreased dose-dependently in SI-treated patients [201]. However, in young healthy subjects, 10-day administration of 2000 mg SI every 12 h also resulted in higher SI levels in blood and CSF with no adverse effects [202]. Further concerns about SI safety have led to the development of SI derivatives, aiming to increase brain penetration, allowing for better efficacy with lower doses of SI for treating AD. A recent report has highlighted AAD-66, a guanidine-appended SI derivative that has improved cognitive performance in a 5xFAD mouse model of AD, concomitant with a reduction in Aβ deposition and glial reactivity as compared to free SI when both were administered ad libitum at the same dose [203].

Although high doses of SI have not been approved for clinical use due to the adverse effects, the search for inositol derivatives, especially those from SI, represent a promising reality in the use of substances derived from natural compounds for the treatment of AD. Along with SI, DPIN is also being tested in clinical trials for AD pathology treatments. DPIN has shown to be a γ-secretase inhibitor that is Notch-sparing (that is, it does not affect Notch cleavage) in vitro [204]. These results show that inositols can directly intervene in the Aβ pathology of different targets (Figure 4).

#### 6.1.2. Inositol Use for Brain Insulin Resistance in Alzheimer’s Disease

Insulin resistance has emerged as one cause–effect of AD [205]. Recent studies have shown that cognitive impairment and AD progression are related to a dysfunction in insulin signaling in the hippocampus and frontal cortex. Postmortem analysis of human hippocampal tissue shows a correlation between high serine-inhibitory phosphorylation of IRS1 and oligomeric Aβ plaques, which were negatively associated with working memory and episodic memory [206]. The same study observed that GSK3-β activity was correlated with insulin resistance and tau hyperphosphorylation [206]. Further studies have shown that early hyperactivation of insulin signaling may cause negative feedback mediated by mTOR and decreased biliverdin-A reductase (BVR-A) activity, an oxidative stress-sensitive antioxidant enzyme and second messenger in insulin signaling, controlling IRS1 and Akt serine phosphorylation [207,208]. Brain insulin resistance and increased oxidative stress lead to overall carbonyl and peroxynitrite protein modifications, leading to signaling dysfunction and a decrease in cognitive performance [209]. An in vitro study showed that the accumulation of ADDLs caused a loss of surface IRs, and ADDL-induced oxidative stress and synaptic spine deterioration could be completely prevented by insulin treatment [209].

Since insulin resistance is an accepted contributor to a worsening AD condition, some strategies have been designed in order to restore insulin signaling in the brain. A proposed strategy is the use of intranasal insulin [210], as it has been proven to be effective at restoring cognitive function, decreasing Aβ aggregation, tau hyperphosphorylation, and nitrosative stress in a 3xTg mouse model of AD [211]. Intranasal insulin has been part of clinical trials, with minimal safety concerns reported so far [212]. Moreover, insulin is a short-life acting molecule, and some derivatives are currently under development [212,213]. Although it is beyond the scope of this review to focus on the insulin molecule itself as a potential treatment for brain insulin resistance-related pathologies, this short summary paves the way for the use of naturally occurring insulin-mimetic compounds, the safety of which has been tested in humans.

The potent stimulatory effect of DCI on insulin signaling is highly likely to contribute to an insulin neuroprotective effect on AD. DCI is effectively transported through the blood–brain barrier and stimulates insulin signaling, as seen in the hypothalamus [214]. An in vitro study showed that DCI, its methyl derivative DPIN, and DCI-GPI INS-2 increased IR autophosphorylation at a dose of 100 μM in primary rat hippocampal neurons [107]. Moreover, the same dose of DCI potentiated insulin-mediated inhibition of ADDL binding to neuron spines and neurites and ADDL-induced synapse damage to neurons [107]. This effect was not observed for MI and was suppressed after addition to PI3K, ERKm, and IGF1R inhibitors, blocking insulin DCI-potentiated signaling [107]. Although these results suggest inositols such as DCI or DPIN are an effective treatment for preserving insulin-deficient signaling in the brain, there are few data on the use of insulin-mimetic inositols in vivo. This is likely due to biased use of SI amongst all inositol derivatives, given its direct interaction with Aβ aggregates.

The use of strong insulin-sensitizers like DCI and their derivatives could serve as an alternative treatment, based on compounds easily obtained from natural sources, whose use in high doses in humans has proven to be safe and effective in other pathologies caused by insulin resistance [27,215]. Moreover, increasing evidence suggesting that peripheral type 2 diabetes exacerbates AD development raises the interest for combinational therapies. The possible protective role of inositols in T2D and AD comorbidity will be summarized in the next section.

#### 6.1.3. Unhealthy Dietary Habits and Microvascular Damage in Alzheimer’s Disease: Preventive Inositol Supplementation

Since brain insulin resistance contributes to AD development, increasing evidence suggests that peripheral type 2 diabetes mellitus (T2DM) may overlap and exacerbate AD-related cognitive impairment, neuroinflammation, oxidative stress, Aβ aggregation, tau hyperphosphorylation, and synaptic dysfunction [216,217,218]. A meta-analysis has shown a 56% increased risk of AD in diabetic patients, and a high prevalence of mixed pathologies [219].

A high fat diet in experimental models, leading to development of T2DM, produces Aβ deposition through altered mechanisms of autophagy and apoptosis, as well as neuroinflammation though alteration in the metabolism and the production of ROS and pro-inflammatory mediators [220,221,222,223]. Middle-aged patients with insulin resistance share common features with AD as the uncoupling of macrovascular blood flow and microvascular perfusion, which is likely due to coupling through the metabolic alterations derived from metabolic shifts induced by the oxidation of fatty acids [224,225]. In addition, high fat diets can modify microbiota compositions, altering the reaction of the intestinal immune barrier. These events might result in changes in circulating levels of pro-inflammatory mediators (cytokines, chemokines, endotoxin) produced at the intestinal levels [226].

The concomitant combination of insulin resistance, a chronic pro-inflammatory state, oxidative stress, and vascular endothelial dysfunction might directly promote an adverse environment for neuronal survival in the context of AD, thus worsening AD-related features. As we have previously described, inositols have been effectively used for treatment of insulin resistance-related pathologies. Regarding insulin-based therapies in AD, the use of intranasal insulin has no cognitive benefits in prediabetic animals compared to non-diabetic animals [227].

In this perspective, the fact that inositols exert an insulin sensitizing effect, but also directly improve endothelial function and act as antioxidant molecules, suggests its use as a supplement in a preventive way. Although no publication exists on the use of Alzheimer’s-associated inositols in the context of insulin resistance, some patents have covered this issue. Pasinetti showed that treatment with 100 mg/kg DPIN administered ad libitum in a Tg2576 mouse model of AD exposed to a high fat diet reduced Aβ levels in the hippocampus, neocortex, and serum through the restoration in the brain of insulin receptor signal transduction [204].

Because inositols are insulin-sensitizers that restore deficient insulin signaling without hyperactivation of insulin signaling, avoiding insulin side-effects, their preventive use as prophylactic agents has emerged as a powerful strategy to delay or lessen the impact of cognitive decline, protecting from synaptic dysfunction. Therefore, we suggest a combination of inositol supplementation with healthy dietary intervention, which is able to modulate microbial production of inositols like SI for the pre-treatment and treatment of AD pathology [228].

### 6.2. Down’s Syndrome

DS is a high-risk factor for the early development of AD. Cognitive impairment in adults with DS resembles that of AD patients [229]. As observed in AD, adults with DS present greater brain MI levels, increasing with age [156]. This may be due to an extra copy of the SMIT1 gene in chromosome 21 [230]. Moreover, the overexpression of amyloid precursor APP due to its presence on chromosome 21 is likely the cause for the increased production of Aβ fragments [231]. Brain insulin resistance is also observed to be developed early in DS subjects, as seen in the inhibition of IRS1 and the uncoupling of downstream elements of insulin signaling [232,233]. These alterations are a major factor for triggering early AD in DS.

Given the prevalence of AD in DS, SI (ELND005) has been tested in a phase II clinical trial in subjects with DS. Both doses of 250 mg/day and 250 mg/twice a day of SI resulted in no serious adverse events, and increased SI concentration in plasma. Interestingly, although a small sample group was tested, the subjects with double dose of 250 mg SI showed improvements in their “neuropsychiatric inventory”, a behavioral measure that assesses symptoms like irritability and agitation [234]. A phase III clinical trial with SI in DS is expected to start.

### 6.3. Anxiety, Compulsive, and Depressive Disorders

Anxiety or closely related disorders are characterized by fear and anxiety related to behavioral disturbances, including panic disorder or obsessive-compulsive disorder (OCD). The search for new treatments for anxiety disorders has long led to an analysis of the effect of MI administration. It has been shown by ^1^H-MRS that frontal cortex MI levels are reduced in patients with depressive and sleep disorders, and MI levels are negatively correlated with depression severity [235,236]. Despite a mechanism of action that is partially unknown, many clinical trials in humans have probed MI at a dose ranging from 12–18 g/day, finding that the dosage is effective for the treatment of panic disorders, OCD, and depression, which are included in the broad spectrum of disorders mainly treated with selective serotonin reuptake inhibitors (SSRIs) [20,21,22].

Since serotonin receptors (5-HTs) are G-protein-coupled receptors, the net effect of MI was thought to be mediated by the replenishment of PI and PIP_2_ availability, avoiding receptor desensitization [237]. In a forced swim test in rats, MI administration reduced depressive behavior, which was abolished by the addition of the specific 5-HT_2A_/5-HT_2C_ antagonist ritanserin, but not the 5-HT_1A_/5-HT_1B_/β adrenergic antagonist pindolol, which suggests that the MI anti-depressant mechanism specifically involves 5-HT_2_ subtype receptors [238]. Moreover, the administration of 1.2 g/kg MI for 12 weeks significantly increased type 2 dopamine (D_2_) and slightly increased 5-HT_2_ receptor density in the striatum of guinea pigs, a region involved in locomotor and stereotype behavior [239]. Although the MI mechanism has not been fully elucidated, its use for anxiety and depressive disorders is still under evaluation. It should also be noted that MI has no additive effect with SSRI treatment, which suggests that somehow both families of compounds intersect regarding their mechanism of action [240,241].

## 7. From Patents to Clinical Trials of Inositols

The industrialized market for nutritional products, such as dietary supplements, has experienced spectacular growth during the late 1990s and the last decade, and has been focused on the development of functional and nutraceutical food products. One of the reasons for the expansion has been the change in the trend of consumers towards the use of supplements with a natural origin. Because there is no worldwide legislation that establishes a precise definition of nutraceuticals, their protection and marketing have both raised certain doubts. Generally, an updated term for nutraceuticals is a product of natural origin that possesses study-based beneficial effects on health.

As we have described throughout this review, inositols fall within these characteristics. The fact that they are molecules with multiple beneficial functions for health (i.e., via their antioxidant capacity, improvement of insulin resistance, and treatment of cardiovascular problems) has led to a high competitiveness to patent their use in an ambit where effective treatments are not available. This is the figure associated with the Alzheimer’s market. Since the mid-2000s, there has been a significant increase in the number of patents related to the use of inositols in AD. To document this, we have performed an advanced search on the World Intellectual Property Organization (WIPO) database, combining results for all nine inositol stereoisomers and derivatives, excluding phosphate-linked forms of inositols and Alzheimer’s disease. After compiling all the results that contained the terms described in the title, abstract, or claims, we carried out a refinement process (elimination of duplicates, exclusion of “inositol derivatives” if not specified) and obtained a total of 26 patents distributed according to the mentioned compounds (Figure 5).

In this race to patent publication, however, it should be kept in mind that while some justify the substantiated use of a specific compound, others rely on general formulations of inositols. On the other hand, some of the studies that support the claims include AD either as the main objective in a set of common diseases, or secondarily due to the overlap of previous patents or the lack of consistency for results in an AD model. Despite the tendency slowing, the number of patents has continued to increase during 2020s, with the search for functional radical-substituted inositol derivatives prioritized over inositol stereoisomers. Although not depicted, many patents have emerged regarding the use of inositols as “carriers” of other substances, allowing them to access targets in the brain.

The late goal of nutraceutical compounds is to be approved in clinical trials. The importance of this resides in the definition of an adequate dose and the approval for use as adjuvant therapies, as in the case of AD. Since nutraceuticals are generally marketed for safe human use, this results in an advantage over synthetic drugs. From 2007 on, at least five phase II clinical trials for SI and two phase II clinical trials for DPIN have been completed for AD and AD-like dementia (Table 3). As mentioned earlier, some concerns have arisen regarding high dosages of SI. Nevertheless, a 250 mg dose of SI results in some significant improvements in the biochemical and cognitive parameters of subjects. Moreover, given the outcome of SI in prevention of AD in DS patients, it is expected that a phase III trial will begin, as the phase II trial mentioned here only enrolled 23 participants. It should also be expected that clinical trials for DPIN and DCI will be extended in the following years, as these are the most promising candidates for the palliation of AD development.

## 8. Concluding Remarks

In this work, we have assessed the use of nutraceuticals such as inositols, which are mostly components of human organisms and are generally acquired through diet. The nutraceutical industry is booming due to the change in the perception of pharmacological drugs and the search for alternatives based on natural compounds. These molecules can be derived from extraction processes in plants or fermentation in bioreactors, which is an advantage over chemical synthesis processes.

Inositols, essentially MI and DCI, are usually a part of membrane structures like phospholipids. As such, the study of inositols has been instrumental in understanding the mechanism of insulin resistance. At a given concentration, inositol stereoisomers can stimulate insulin signaling, preferably as IGs, but also show antioxidant capacity. However, after revising the proposed models for inositols in insulin signaling, it is still uncertain whether inositols per se are capable of displaying these characteristics. In addition to supporting insulin signaling, the importance of inositols in the brain appears to lie in their ability to control neuron and glia responses to external environments. In this regard, psychiatric disorders such as depression and anxiety were associated with imbalances in MI content in brain. Treatment with MI has been shown to improve behavioral outcomes in depression, anxiety, panic, and obsessive-compulsive disorders in a mechanism of action related to serotonin receptor activation. Albeit not fully elucidated, MI role may imply PI availability, standing out the importance of MI in physiological regulation of serotonin signaling in behavior.

Inositols are solid candidates for developing new approaches for the treatment or improvement of chronic diseases associated with aging, especially those linked to insulin resistance and oxidative stress. Aging of the population is a challenge for developed countries due to the increasing incidence of dementia. Amongst all types of dementia, sporadic AD represents a heavy burden for patients, their families, and society. AD treatment is considered a high risk, high reward market. Different strategies have been approached in an attempt to merge effective therapy when the disease is clinically diagnosable. This process generally renders unsuccessful results due to the complexity and partial lack of knowledge of the progression of AD development. We have approached AD from the point of view of the initial stages of the disease, as the aging process is closely related to the development of metabolic dysregulations leading to insulin resistance, where inositols might have a potential beneficial use.

This exhaustive review of the physiological role of inositols helps us to understand what benefits can result from external administration, either to compensate for their deficiency or as an adjuvant in oxidative stress processes. The specific position of the hydroxyl radicals gives certain stereoisomers such as SI or EI specific characteristic to limit the aggregation of Aβ, one of the main pathophysiological characteristics of AD and cognitive dementia observed in DS.

The different inositols offer complementary strategies for the preventive treatment of AD. Research of inositols, especially those with a remarkable insulin-sensitizing capacity, such as DCI or its derivative DPIN, may offer a different perspective on how insulin resistance develops in the brain and its contribution to improving cognitive outcomes for AD. The fact that clinical studies on neurological disorders with inositols have been carried out in the last decade meets a necessity and opens a way to the expansion in the use of inositols, whose perspective is not that of being a therapeutic replacement, but a bioactive compound that helps to prevent or suppress decline in neurodegenerative diseases.

## Figures and Tables

**Figure 1 biomedicines-08-00295-f001:**
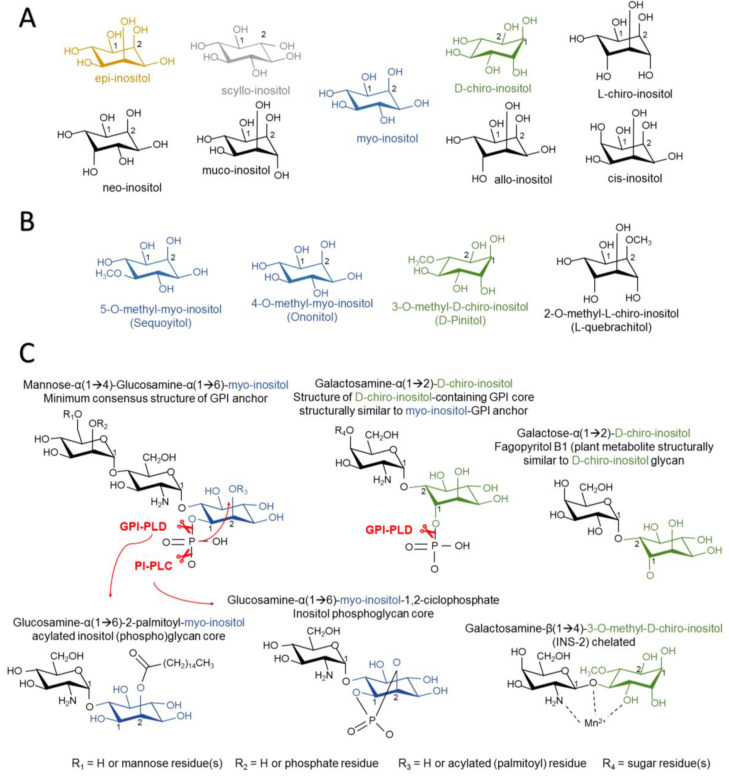
Structure in the chair conformation of inositol stereoisomers (**A**), inositol methyl derivatives (**B**), natural and synthetic inositol phosphoglycan cores, and insulin-mimicking inositol phosphoglycans (**C**). Glycophosphatidyl inositol phospholipase D (GPI-PLC) hydrolyzes phosphate inositol-lipid linkage, releasing unphosphorylated inositol. Phosphatidylinositol phospholipase C (PI-PLC) hydrolyzes phosphate inositol-lipid linkage in α(1→6) myo-inositol (but not α(1→2) d-chiro-inositol) structures when the C2 position is not occupied by an acyl-lipid chain, promoting cyclic (1,2) phosphate linkage to myo-inositol.

**Figure 2 biomedicines-08-00295-f002:**
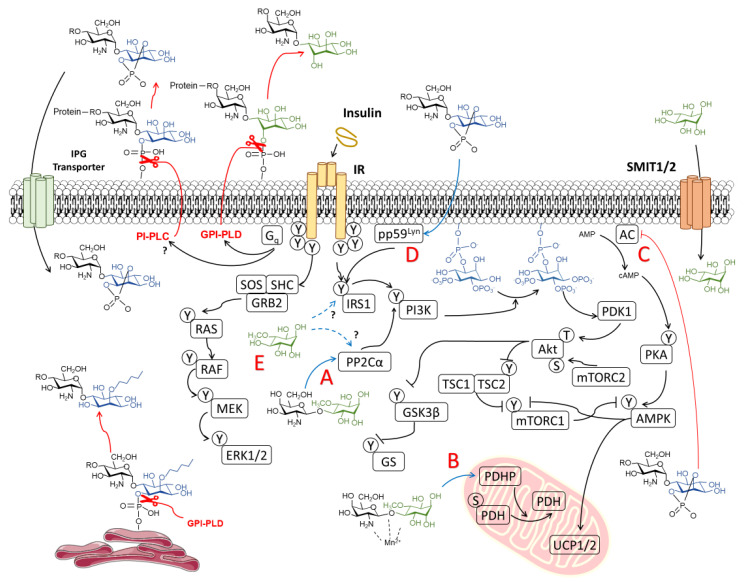
Proposed mechanism of action of inositols in insulin signaling. Non-canonical insulin signaling through the G_q_ protein presumably stimulates glycophosphatidylinositol phospholipase D (GPI-PLD) and/or (glycol)phosphatidylinositol phospholipase C (PI-PLC), mediating the hydrolysis of phosphate linkage between inositol and membrane lipids, leading to the release of inositol phosphoglycans (IPGs). Acylated-IPGs are formed in the plasma membrane and endoplasmic reticulum. IPGs are internalized via an IPG transporter. Insulin-sensitizing properties of inositols correspond to (**A**) allosteric modulation of protein phosphatase 1A (PP2Cα) and (**B**) pyruvate dehydrogenase phosphatase (PDHP), as observed by d-chiro-inositol glycan (INS-2), and (**C**) the inhibition of adenylate cyclase (AC) and protein kinase A (PKA) activity (observed with myo-inositol glycans). IPGs may also upregulate IRS1 signaling by (**D**) activating the upstream modulator pp59^Lyn^ localized in lipid rafts. Free inositols such as d-pinitol also exert (**E**) insulin-mimetic properties in the absence of an insulin stimulus.

**Figure 3 biomedicines-08-00295-f003:**
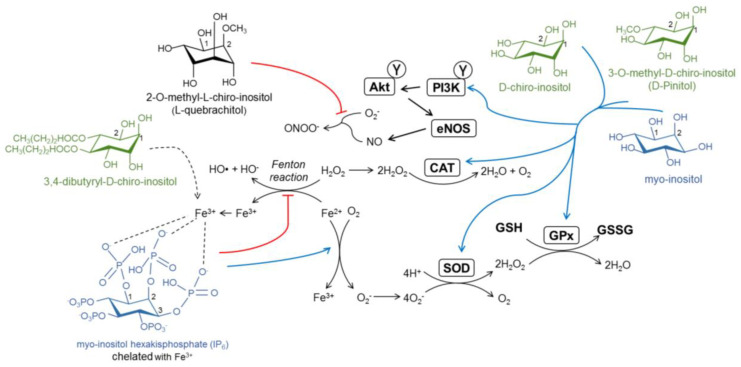
Proposed antioxidant mechanisms of inositol derivatives. Phytic acid (IP_6_) chelates and sequesters Fe^3+^, redirecting Fe^2+^ via the Fenton reaction to oxidation with O_2._ Four molecules of Fe^2+^ are necessary to oxidize one molecule of O_2_, which generates O_2_^−^, which is later converted in H_2_O_2_, and finally inactive H_2_O. Reactive oxygen species (ROS) are scavenged by the antioxidant enzymes superoxide dismutase (SOD), catalase (CAT), and glutathione peroxidase (GPx). Inositols upregulate antioxidant enzyme levels and activity. The activation of insulin signaling in the endothelium results in increased NO production. l-quebrachitol and 3,4-dibutyryl-d-chiro-inositol scavenge peroxynitrite (ONOO^−^) and ROS species, respectively, although the mechanisms of this have not been unveiled yet.

**Figure 4 biomedicines-08-00295-f004:**
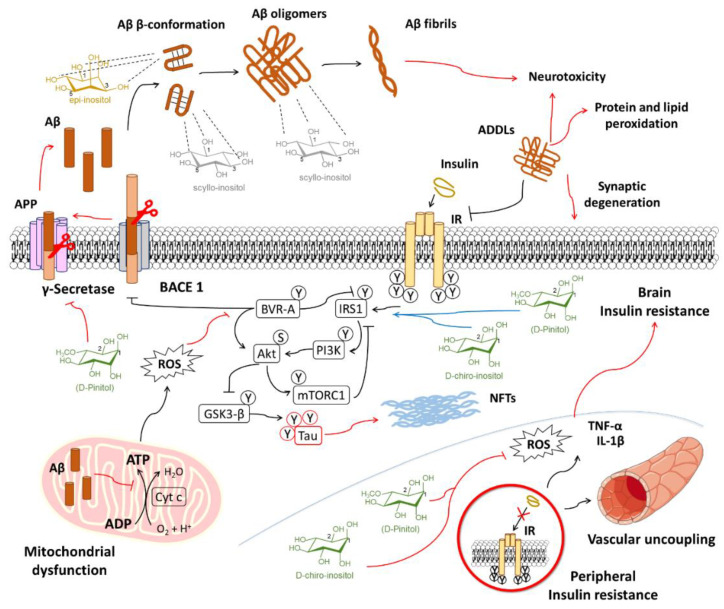
Schematic view of inositol interplay in mechanisms leading to progression of Alzheimer’s disease. Epi- and scyllo-inositol stabilize β-sheet conformation of β amyloid (Aβ) products of aberrant APP hydrolysis by β-secretase (BACE1) and γ-secretase. Scyllo-inositol also prevents fibril formation from Aβ oligomers. Pinitol is seen to inhibit γ-secretase cleavage of APP but not Notch products. Brain insulin resistance contributes to amyloid pathology and ultimately to tau hyperphosphorylation, forming toxic microtubule neurofibrillary tangles (NFTs). Mitochondrial dysfunction and presence of Aβ-derived diffusible ligands (ADDLs) worsen the insulin response. d-chiro-inositol and d-pinitol improve insulin sensitivity in the brain, counteracting the development of brain insulin resistance. Inositols prevent early impairment in insulin signaling and the development of vascular dysfunction, which promotes oxidative stress and the release of pro-inflammatory cytokines, ultimately contributing to amyloid pathology in the context of brain insulin resistance in Alzheimer’s disease.

**Figure 5 biomedicines-08-00295-f005:**
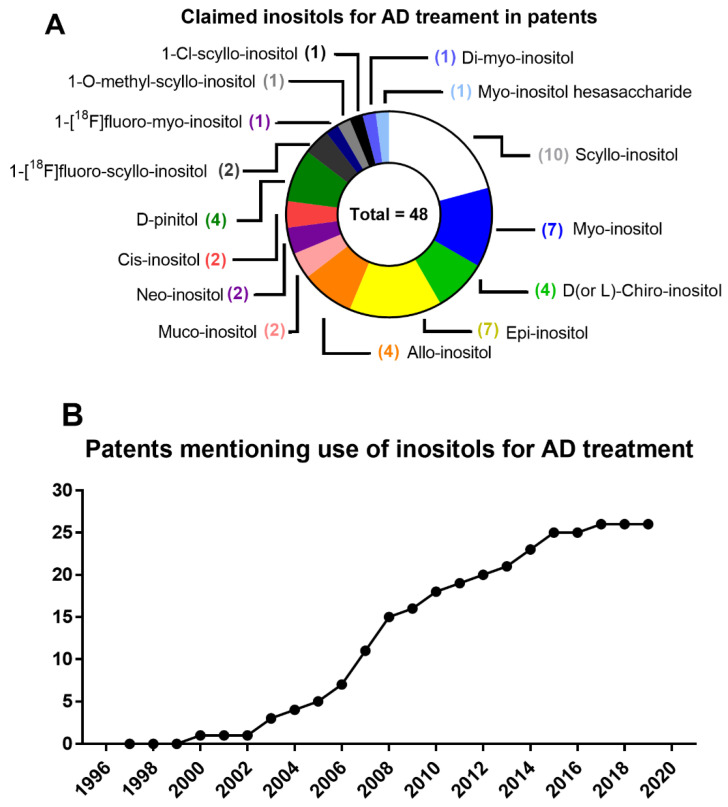
(**A**) Graphical illustration of patents claiming the use of inositols for Alzheimer’s disease, including patents mentioning more than one inositol stereoisomer and/or derivative. (**B**) Number of patents mentioning and claiming use of inositols for AD treatment in the last two decades.

**Table 1 biomedicines-08-00295-t001:** Biological sources of inositol stereoisomers, distribution in mammalian tissue, and current pharmacological applications for the treatment of human diseases. PIs: Phosphatidylinositols; IP_3_: Inositol 1,4,5-trisphosphate; GPIs: Glycosylphosphatidylinositols.

Inositol Stereoisomer	Mainly Found in	Distribution and Function in Mammals	Pharmacological Properties
Myo-inositol	Plants Mammals Bacteria	Found in all tissues. Part of membrane PIs. MI in PIP_2_ and PIP_3_ acts in several intracellular signaling pathways. MI in IP_3_ regulates metabolic flux, intracellular Ca^2+^, and membrane excitability. Acts as an osmolyte.	Treatment of polycystic ovary syndrome (PCOS) related to insulin resistance and hyperandrogenism in combination with DCI [6,7]. Prevention of gestational diabetes mellitus in combination with DCI [19]. Treatment of obsessive-compulsive disorder, panic disorder, and depression [20,21,22].
Epi-inositol	Plants	Found in brain, liver, muscle, blood, and other tissues, epimerized from myo-inositol [13]. Biological function unknown.	Decreases the aggregation of toxic forms of β-amyloid protein in Alzheimer’s disease [23,24].
Scyllo-inositol	Plants Mammals Bacteria	Found preeminently in the brain [12,13,25]. Acts as an osmolyte. Competes with MI for active transport into the cell.	Decreases the aggregation of toxic forms of β-amyloid protein in Alzheimer’s disease [23,24]. Inhibits the aggregation of α-synuclein in Parkinson’s disease [26].
Neo-inositol	Plants	Detected in brain [12]. Biological function unknown.	No pharmacological properties described.
Muco-inositol	Plants	Found in brain, liver, muscle, blood, and other tissues, epimerized from myo-inositol [13]. Biological function unknown.	No pharmacological properties described.
d-chiro-inositol	Plants Mammals Bacteria	Mainly found in insulin-responsive tissues. Acts as a second messenger in insulin signaling pathway, likely as a part of GPIs.	Treatment of polycystic ovary syndrome (PCOS) related to insulin resistance and hyperandrogenism in combination with MI [6,7]. Prevention of gestational diabetes mellitus in combination with MI [27].
l-chiro-inositol	Plants	Not detected in mammalian tissue.	No pharmacological properties described.
Allo-inositol	Plants Mammals Bacteria	Not detected in mammalian tissue.	Non-potent inhibitor of the aggregation of toxic forms of β-amyloid protein in Alzheimer’s disease [28]
Cis-inositol	Non-naturally occurring	Not detected in mammalian tissue.	No pharmacological properties described.

**Table 2 biomedicines-08-00295-t002:** Distribution of inositols in the brain and the main functions of inositol derivatives. LPI: Lysophosphatidylinositol; GPR55: Cannabinoid G protein-coupled receptor 55; HMIT: H^+^/myo-inositol transporter; PIP_2_: Phosphatidylinositol (4,5)-bisphosphate; GIRK: G protein-gated inwardly rectifying potassium.

Inositol Sources in the Brain	Inositol Isoform	Main Brain Target	Described Mechanism
De novo synthesis from active transport from peripheral tissues. SMIT1: A 2 Na^+^:1 myo-inositol co-transporter, present in microglia and neurons [55]. Expressed in the cerebellum, hippocampus, and cortex [47]. SMIT2: A 2 Na^+^:1 myo-inositol co-transporter predominantly present in neurons [55]. Expressed in the cerebellum, hippocampus, and cortex [47]. HMIT: H^+^/myo-inositol co-transporter, present in neurons and microglia [52,54]. Expression found in the cerebral cortex, hippocampus, hypothalamus, cerebellum, and brainstem. Participates in the regulation of intracellular IP3 levels [52,54].	Myo-inositol	Astrocytes.	Acts as an osmolyte in astrocytes. Alterations in brain MI levels are commonly observed in traumatic and neurodegenerative diseases [154,155,156,157].
	LPI	GPR55 in the hippocampus.	Binding of LPI to GPR55 enhances long-term potentiation and stimulates memory formation [151,152] via the mobilization of intracellular Ca^2+^ [153].
	IP_3_	Endoplasmic reticulum in neurons.	Mobilization of intracellular Ca^2+^ deposits in neurons, modulating membrane excitability in a mechanism associated with HMIT transport [54].
	PIP_2_	Potassium channels GIRK and KCNQ2/3, mainly found in the hippocampus.	PIP_2_ bioavailability modulates neuronal excitability in a SMIT1-KCNQ2/3-dependent mechanism [158]. PIP_2_ stimulates the opening of GIRK and KCNQ2/3 channels, promoting M-current [159,160].
	PI(3)P	GABA neurons in the hippocampus.	Promotes postsynaptic GABA receptor clustering [161].
	PI(4,5)P_2_	Ion channels found in neurons.	Regulation of neuronal excitability, as a substrate for normal function of ion channels in neurons [146,162].
	PI(3,5)P_2_	NMDA and GluA1 channels in neurons.	Altered PI(3,5)P_2_ impairs recycling of NMDA and GluA1 channels, affecting glutamatergic transmission [163,164].
	PI(3,4)P_2_	Neurites.	PI(3,4)P_2_ clustering is necessary for neurite initiation [165].

**Table 3 biomedicines-08-00295-t003:** Clinical trials of scyllo-inositol and d-pinitol in Alzheimer’s disease-related patients. Alzheimer’s Disease Cooperative Study-Activities of Daily Living (ADCS-ADL), Alzheimer’s Disease Cooperative Study Clinician’s Global Impression of Change (ADCS-CCGIC), Alzheimer’s Disease Assessment Scale-Cognitive Subscale (ADAS-Cog), Clinical Dementia Rating-Sum of Boxes (CDR-SB), Mini-Mental State Examination (MMSE), Neuropsychiatric Inventory (NPI), NPI-C Combined Agitation and Aggression Subscores (NPI-C A+A), Neuropsychological Test Battery (NTB), Treatment Emergent Adverse Events (TEAEs).

Inositol	Title, NCT Number, and Date	Dose	Population	Outcome Measures	Published Results
Scyllo-inositol (ELND005)	ELND005 in Patients with Mild to Moderate Alzheimer’s Disease Study Start: December 2007 Study Completion: May 2010 NCT00568776	Placebo, 250 mg/kg; 1000 mg/kg; 2000 mg/kg at 78 weeks	Enrollment: 353 Age: 50 Years to 85 Years (Adult, Older Adult)	ADCS-ADL ADAS-Cog CDR-SB NPI NTB	[201]
	ELND005 Long-Term Follow-up Study in Subjects With Alzheimer’s Disease Study Start: June 2009 Study Completion: June 2011 NCT00934050	Placebo, 250 mg/kg; post-Dec 2009, 2000 mg/kg; pre-Dec 2009, 48 weeks	Enrollment: 103 Age: Child, Adult, Older Adult	TEAEs	Confidence agreement
	Efficacy and Safety Study of ELND005 as a Treatment for Agitation and Aggression in Alzheimer’s Disease Study Start: November 2012 Study Completion: May 2015 NCT01735630	Placebo, film coated tablets, twice a day for 12 weeks	Enrollment: 350 Age: 50 Years to 95 Years (Adult, Older Adult)	ADCS-ADL ADCS-CGIC MMSE NPI-C A+A	Confidence agreement
	A 36-Week Safety Extension Study of ELND005 as a Treatment for Agitation and Aggression in Alzheimer’s Disease Study Start: January 2013 Study Completion: August 2015 NCT01766336	Continue at same dose as previous clinical trial for 36 weeks	Enrollment: 296 Age: 50 Years to 95 Years (Adult, Older Adult)	TEAEs	Confidence agreement
	A 4-Week Safety Study of Oral ELND005 in Young Adults With Down Syndrome Without Dementia Study Start: September 2013 Study Completion: June 2014 NCT01791725	Placebo, 250 mg/kg once a day; 250 mg/kg twice a day, 4 weeks	Enrollment: 23 Age: 18 Years to 45 Years (Adult)	TEAEs	[234]
d-pinitol (NIC5-15)	Development of NIC5-15 in The Treatment of Alzheimer’s Disease Study Start: January 2007 Study Completion: March 2010 NCT00470418	Placebo; dose not specified, 7 weeks	Enrollment: 15 Age: Child, Adult, Older Adult	TEAEs Clinical Measures of Cognition at Terminal Visit	[242]
	A Single Site, Randomized, Double-blind, Placebo Controlled Trial of NIC5-15 in Subjects With Alzheimer’s Disease Study Start: April 2007 Study Completion: June 2014 NCT01928420	Placebo; dose not specified, 24 weeks	Enrollment: 30 Age: 40 Years to 95 Years (Adult, Older Adult)	ADCS-ADL ADCS-CGIC ADAS-Cog MMSE NPI Changes in AD biomarkers, APO-E genotyping	N/A

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
