# Peer review of "The Biomedical Uses of Inositols: A Nutraceutical Approach to Metabolic Dysfunction in Aging and Neurodegenerative Diseases"

_biomedicines, 2020, doi:10.3390/biomedicines8090295_

Round 1

Reviewer 1 Report

This manuscript entitled "The biomedical uses of inositols: a nutraceutical approach to metabolic dysfunction in aging and neurodegenerative diseases" by  López-Gambero et al. is well written review paper. I am very much impressed by their quality of work. I recommend to accept this manuscript after minor revision.

1. The manuscript is very well written and contains detail information about Inositol and its isomers. It would be better if the details like source, distribution, properties, pharmacological properties in a table.
2. In section 5 Inositol distribution in brain and their activity has been described very widely. For readers benefit and for well understanding, it can be suggested to include one table for organizing all these details.

Reviewer 2 Report

Review of the manuscript “The biomedical uses of inositols: a nutraceutical approach to metabolic dysfunction in aging and neurodegenerative diseases” by Antonio López-Gambero submitted to “Biomedicine”’

Currently aging and related neurodegenerative diseases are an enormous burden requiring huge funds and efforts from society and medical community.  The aging is connected with physical deterioration and weakening of metabolic homeostasis. One of the important signaling pathways altered in aging and neurodegenerative diseases is associated with physiological regulation of insulin signaling. The authors investigated the role of inositols - second messengers involved in the regulation of several cell signaling processes- in insulin signaling.  This is an important field in the study of mechanism of aging and neurodegenerative processes and the results, concepts and hypothesis presented in the review will be interesting for the readership of “Biomedicine”.   

The following corrections should be made:

Abstract

  • Line 31: “The use of natural compounds such as inositols may represent a paradigm in the industrial approach…” The sense of this sentence is unclear. What the authors mean by “industrial approach?” Clarifications needed.
  • There is a contradiction between the title containing “aging” and Abstract and keywords, where “aging” is not mentioned.

Introduction

Lines 94-95: “Central insulin resistance is a common feature linked to premature aging and observed in neurological disorders, including early stages of Alzheimer’s Disease (AD) or Down’s Syndrome (DS).”

The authors should add here a citation to the following article:”Caveolin: A New Link Between Diabetes and AD. Cell Mol Neurobiol. 2020 Jan 23”. doi: 10.1007/s10571-020-00796-4

Line 97 :” Taking in consideration these numbers…” should be replaced by :” Taking in consideration this trend…”

Line 99 :” By 2050, more than 131.5 million people are expected to be affected. Alzheimer’s Disease (AD) leads…”. Alzheimer’s disease was already abbreviated on line 94.

Lines 105-108: ”Up to date results suggest that unhealthy dietary habits, microbiota changes and oxidative stress favor the development of brain insulin resistance which could contribute to a neuroinflammatory profile directly activating both resident immune cells of the brain (microglia),and astrocytes, promoting an adverse environment for neuronal survival in the context of AD”.

The authors should add a citation after this.

Lines 151-152:” inositols are part of glycosylphosphatidylinositols (GPIs)…”. It is not clear what the authors want to say by this statement. Clarification needed.

Line 210:”Transport of inositol isomers in into cells is regulated by…” should be corrected as follows:”Transport of inositol isomers into cells is regulated by…”

Lines 236-238: ”A further look about inositol transport in the brain will be discussed in further sections of this review” should be corrected as : ”A more detailed mechanism of inositol transport in the brain will be discussed in other sections of this review”.

Legend to Figure 1. “Structure in chair conformation of inositol stereoisomers (A), inositol methyl-derivatives (B) and described natural and synthetic inositol phosphoglycan cores and insulin-mimicking inositol phosphoglycans (C)” should be rewritten as “Structure in chair conformation of inositol stereoisomers (A), inositol methyl-derivatives (B), natural and synthetic inositol phosphoglycan cores nd insulin-mimicking inositol phosphoglycans (C).”

Chapter 3.1.1 “Canonical insulin signaling” is overloaded by many details and should be truncated.

Lines 427-428: ”A recent study performed by our group has shown that administration of DPIN under fasting conditions on rats promote a significant reduction of circulating insulin, without affecting plasma glucose levels [91]” should be rewritten as follows: ” Our recent study has shown that administration of DPIN to fasting rats promotes a significant reduction of circulating insulin, without affecting plasma glucose levels [91].”

Chapter 6.2.2 “Inositol use for brain insulin resistance in Alzheimer’s Disease” contains too much materials and should be truncated.